# Integrated analysis of genome-wide association studies and 3D epigenomic characteristics reveal the *BMP2* gene regulating loin muscle depth in Yorkshire pigs

**Yuanxin Miao**[1,2☯], **Yunxia Zhao**[1☯], **Siqi Wan**[1☯], **Quanshun Mei**[1], **Heng Wang**[1], **Chuanke Fu**[1], **Xinyun Li**[1,3], **Shuhong Zhao**[1,3], **Xuewen Xu**[1]\*, **Tao Xiang**[1]\*

**1** Key Laboratory of Agricultural Animal Genetics, Breeding and Reproduction of Ministry of Education & Key Laboratory of Swine Genetics and Breeding of Ministry of Agriculture, Huazhong Agricultural University, Wuhan 430070, China, **2** Research Institute of Agricultural Biotechnology, Jingchu University of Technology, Jingmen 448000, China, **3** Hubei Hongshan laboratory, Huazhong Agricultural University, Wuhan 430070, China

☯ These authors contributed equally to this work.

\* Xuewen_Xu@mail.hzau.edu.cn (XX); Tao.Xiang@mail.hzau.edu.cn (TX)

**Data Availability Statement:** The data of phenotype and genotype are deposited in supporting information files S1 Data and S2 Data.

## Abstract

### Background

The lack of integrated analysis of genome-wide association studies (GWAS) and 3D epigenomics restricts a deep understanding of the genetic mechanisms of meat-related traits. With the application of techniques as ChIP-seq and Hi-C, the annotations of *cis*-regulatory elements in the pig genome have been established, which offers a new opportunity to elucidate the genetic mechanisms and identify major genetic variants and candidate genes that are significantly associated with important economic traits. Among these traits, loin muscle depth (LMD) is an important one as it impacts the lean meat content. In this study, we integrated *cis*-regulatory elements and genome-wide association studies (GWAS) to identify candidate genes and genetic variants regulating LMD.

### Results

Five single nucleotide polymorphisms (SNPs) located on porcine chromosome 17 were significantly associated with LMD in Yorkshire pigs. A 10 kb quantitative trait locus (QTL) was identified as a candidate functional genomic region through the integration of linkage disequilibrium and linkage analysis (LDLA) and high-throughput chromosome conformation capture (Hi-C) analysis. The *BMP2* gene was identified as a candidate gene for LMD based on the integrated results of GWAS, Hi-C meta-analysis, and *cis*-regulatory element data. The identified QTL region was further verified through target region sequencing. Furthermore, through using dual-luciferase assays and electrophoretic mobility shift assays (EMSA), two SNPs, including SNP rs321846600, located in the enhancer region, and SNP rs1111440035, located in the promoter region, were identified as candidate SNPs that may be functionally related to the LMD.

**Funding:** TX acknowledges funding from the National Key Research and Development Program of China (No. 2021YFF1000601 & 2022YFD1301900), Major Science and Technology Projects in Hubei Province (No.2020ABA016); SZ acknowledges funding from the China Agriculture Research System of MOF and MARA (CARS-35); YM acknowledges funding from Natural Science Foundation in Hubei Province (No. 2022CFB021), the Scientific Research Program Key Project of Hubei Provincial Department of education (D20214301), the Natural Science Foundation of Jingmen City (2022YFZD051), and the research fund from Jingchu University of Technology (ZD202102, 2022ZD005). The funders had no role in study design, data collection and analysis, decision to publish, or preparation of the manuscript.

**Competing interests:** The authors have declared that no competing interests exist.

## Conclusions

Based on the results of GWAS, Hi-C, and *cis*-regulatory elements, the *BMP2* gene was identified as an important candidate gene regulating variation in LMD. The SNPs rs321846600 and rs1111440035 were identified as candidate SNPs that are functionally related to the LMD of Yorkshire pigs. Our results shed light on the advantages of integrating GWAS with 3D epigenomics in identifying candidate genes for quantitative traits. This study is a pioneering work for the identification of candidate genes and related genetic variants regulating one key production trait (LMD) in pigs by integrating genome-wide association studies and 3D epigenomics.

## Author summary

The loin muscle depth (LMD) is positively correlated with lean meat content and can therefore be used to estimate the lean meat content of pigs. The loin muscle depth is a quantitative trait that is influenced by whole-genome distributed polygenes. To identify the possible candidate genes and mutations impacting the LMD, we used genome-wide association study (GWAS), which is commonly used to identify candidate genomic regions associated with complex growth traits, to mapping the genomic regions. As over 90% of the identified variants have been localized to non-coding regions of the pig genome and their functions in phenotype regulation are poorly understood, we further combined 3D epigenomics to understand the mechanism of genetic variants regulating LMD. In combination with different types of data and analyses, we identified a 10 kb quantitative trait locus (QTL) on chromosome 17 that was significantly associated with LMD. We identified the *BMP2* gene as a major candidate gene, and SNP rs1111440035 and rs321846600 were identified as likely candidate mutations affecting the LMD. Our study is unique in its attempt to identify candidate genetic variants by integrating GWAS and 3D epigenomics in pigs.

## Introduction

Lean pigs provide the majority of pork for the consumer market, and lean meat content is a critical breeding goal for the pig industry [1,2]. The loin muscle depth (LMD) is defined as the minimum distance from the vertebral channel to the cranial end of the gluteus medius muscle. The LMD is usually measured by ultrasound, and it is significantly positively correlated with lean meat content. Therefore, selection for LMD enhances the lean meat content of pigs over time. Revealing the genetic mechanisms governing LMD and identifying genes significantly associated with LMD may further enhance the efficiency of LMD improvement.

   Genome-wide association study (GWAS) is a powerful and effective strategy for detecting QTL regions associated with complex traits [2–6]. It has been successfully used to identify candidate genes associated with both LMD and loin muscle area (LMA). LMD is significantly positively correlated with LMA (r = 0.945), and it also can be used to predict lean meat content [7]. For instance, 9 independent genome-wide SNPs accounting for 27.51% of phenotypic variation associated with LMD were identified from 370 Chuying-black pigs using GWAS [8]. Another example has demonstrated that GWAS meta-analysis has identified multiple QTL regions related to LMA or LMD across 6,043 Duroc pigs, and these QTL regions contained 75 significantly associated SNPs [9]. To date, 415 QTLs for loin muscle area and 97 QTLs for loin

muscle depth are available in the Pig QTL Database (https://www.animalgenome.org/cgi-bin/QTLdb/SS/index) [6,10–12].

Although over 90% of these identified variants have been mapped to non-coding regions in the pig genome, there is limited understanding of their functions in these non-coding regions [13]. To uncover the molecular mechanisms of these variants, it is necessary to elucidate not only the genetic variations but also the epigenetic activities accompanying this genetic variation. Previous studies have indicated that integrating GWAS and epigenetic data, including chromatin states [14], open chromatin regions [15], high-throughput chromosome conformation capture (Hi-C) [16], and RNA-seq [17], could enhance our understanding of the identified variants and their target genes as well as the relationships between genotypes and phenotypes. Recently, the accumulation of epigenetic data [18] offers an opportunity to interpret GWAS results from pigs to reveal pertinent genetic mechanisms of specific traits.

This study was aimed at identifying major candidate genes as well as key candidate genetic variants influencing phenotypic variation in LMD and exploring the genetic mechanisms governing LMD regulation in Yorkshire pigs. In this study, we performed a GWAS of LMD in Yorkshire pigs, followed by a chromatin state analysis in a region significantly associated with LMD based on genome-wide high-resolution profiles from ChIP-seq and Hi-C data. Our work provides a foundational framework and a successful case for combining GWAS and 3D epigenomics in pigs.

## Materials and methods

### Ethics statement

All procedures involving tissue samples collection and animal care were performed according to the approved protocols and ARRIVE (Animal Research: Reporting In Vivo Experiments) guidelines and were approved by the Ethics Committee of Huazhong Agricultural University (HZAUSW-2018-008).

### Phenotypic recordings and genotypes

In this study, loin muscle depth (LMD) from the 10[th] rib to 11[th] rib of pigs was measured in Yorkshire pigs with the weight of 100 ± 5 kg by an Aloka 500V SSD B ultrasound. In total, LMD of 16,533 pigs was recorded in the years from March 2014 to January 2018. The pedigrees of these measured pigs could be traced back at least five generations with a total of 42,245 pigs included in these pedigrees. Genomic selection was started late in 2016 and finished at the end of 2018 due to the outbreak of African swine fever. Ear tissues were collected by the criterion of at least 2 males and 2 females in each litter. A total of 2,733 Yorkshire (YY) pigs provided both LMD recordings and ear tissue samples simultaneously.

Total DNA was extracted from these 2,733 pigs and genotyped using the Geneseek Porcine 50K SNP Chip (Neogen, Lincoln, NE, United States), and 50,703 SNP markers across the genome were obtained. SNPs were mapped to the Sscrofa11.1 pig genome assembly. Quality control (QC) was carried out with the following parameters set: individual call rate$\geq$90%; SNP call rate$\geq$90%; minor allele frequency$\geq$0.01; Hardy-Weinberg equilibrium P-value$\geq$10$^{-6}$. After quality control, a total of 42,772 SNPs were obtained from each of 2,733 pigs, and these SNPs were used for subsequent genome-wide association analysis.

### Pre-correction of phenotypes

The loin muscle depth (LMD) of 16,533 pigs was pre-corrected. The single-step GBLUP (ssGBLUP) method, which incorporates both pedigree and genomic information, was used to

 

estimate variance components and genomic estimated breeding values (GEBVs) [19,20]:

$$y = Xb + Za + e \qquad (1)$$

where $y$ is a vector of original phenotypic values of LMD of 16,533 pigs; $Xb$ accounts for the fixed effects of herd-year-season, sex, and age; $a$ is a vector of random additive genetic effects; and $Z$ is the incidence matrix; $e$ is a vector of residual effects, which was assumed to follow normal distribution as $e \sim N(0, I\sigma_e^2)$, where $I$ is an identity matrix. It was assumed that the random additive effects follow a normal distribution $a \sim N(0, H\sigma_a^2)$, where $H$ is a relationship matrix combining pedigree and genomic information, which was constructed by previously reported method [19]. $\sigma_a^2$ is the additive genetic variance. Software DMU [21] was used to estimate the variance components and solve the genomic model. The corrected phenotypes $y_c$ were calculated as the sum of estimated additive genetic effects and the residuals ($y_c = \hat{a} + \hat{e}$) for all the pigs.

## Genome-wide association studies

The genome-wide association study (GWAS) was performed on 2,733 pigs with both phenotypic and genotypic information by using a mixed linear model-based association analysis (MLMA) in software rMVP [22]. The following mixed linear model was used for GWAS:

$$y_c = 1\mu + Xg + Wu + e, \qquad (2)$$

where $y_c$ is a vector of corrected phenotype of LMD in the genotyped 2,733 Yorkshire pigs; $\mu$ is the overall mean of corrected phenotypes; $1$ is a vector of ones; $X$ is a matrix of the SNP genotypes with entry 0, 1, 2 indicating genotype $AA$, $AB$, and $BB$, respectively; $g$ is the fixed additive genetic effect of the analyzed SNP; $u$ is a vector of random polygenic effects, with an assumption that $u \sim N(0, G\sigma_u^2)$, where $G$ is the marker-based additive genomic relationship matrix, which was constructed by the method reported by Vanraden (2008) [23]; $\sigma_u^2$ is the polygenic variance; $W$ is the incidence matrix between the corrected phenotype and the corresponding random polygenic effects; $e$ is a vector of random residual effects, with an assumption that $e \sim N(0, I\sigma_e^2)$. In Bonferroni corrections, the genome-wide significant threshold was set as ($-\log_{10}[0.05/\text{number of SNPs}] = 5.93$).

## Identification of LD block and QTL analysis

Identification of linkage disequilibrium (LD) blocks was performed in the chromosomal regions containing the identified significantly associated SNPs by software Haploview [24]. The QTLs located in the identified LD blocks were searched from Pig QTL database (pigQTLdb, https://www.animalgenome.org/cgi-bin/QTLdb/SS/index).

## Haplotype analysis

Haplotypes were constructed in the LMD-associated region on SSC17 through the Linkage disequilibrium and linkage analysis (LDLA) method [25]. The LDLA method can accurately localize the identified QTL regions by combining the results of populational linkage disequilibrium and within-family linkage, thus LDLA was used for the haplotype analysis [26,27]. The haplotypes were constructed by the software PHASEBOOK [28] using the Hidden Markov Model [28] based on an assumption that there existed a predetermined number of ancestral haplotypes (K = 20), and that all haplotypes in the population were derived from these ancestral haplotypes [29]. A likelihood ratio test (LRT) was performed along chromosomes to test

the presence of a QTL region in the given map [30]. The 95% confidence interval (CI) was calculated as the LRT value of the most significant loci minus 2 [31,32].

## Integrated analysis of pig epigenomics datasets

The topologically associated domains (TAD), sub-domains, Hi-C contact matrix data, significant H3K27ac peaks, and enhancer-gene pairs were all from our previous study based on the Sscrofa11.1 pig genome [18]. Hi-C data were analyzed using HiC-Pro pipeline version 2.9.0 software for genome mapping. The insulation score and top domain methods were used to perform TAD calling and sub-domain identification. ChIP-seq data analysis included reads mapping (BWA v0.7.15), low-quality reads filtering (SAMTools v1.9 and Picard v1.126), and peak calling (MACS2 v2.1.0). The details of Hi-C and ChIP-seq data analysis were described in our previous study [18]. In this study, the Hi-C contact and TAD structure integrated with GWAS were visualized using Juicebox [33]. The significant H3K27ac peaks across various tissues were merged using BEDTools v2.26.0 [34]. Significant H3K27ac peaks surrounding the candidate genes were visualized in the IGV browser [35].

## SNP polymorphism identification and association analysis

To further identify important functional variants, the genomic region between 15.51 Mb and 16.31 Mb on SSC17 was deeply sequenced (>20X) by using target region sequencing technology on 732 randomly selected individuals across the entire population. Target region sequencing libraries were created from isolated DNA according to the manufacturer's instructions. The high-quality libraries were sequenced using the Illumina HiSeq3000 platform, which generated paired-end sequencing data. Quality control of paired-end sequencing reads was conducted with Trimmomatic, followed by alignment to the pig reference genome using BWA. The variants were identified using GATK software according to specific criteria (Qual score $\geq$ 30, QD < 20.0, ReadPosRankSum < $-8.0$, FS > 10.0 and QUAL < \$MEANQUA). Subsequently, SNPs located between 15.51 Mb and 16.31 Mb on SSC17 and distributed within the region of significant H3K27ac peaks were selected for trait association analysis with the corrected LMD phenotypes across the 732 randomly-selected individuals. The association analysis between an individual SNP and LMD was carried out using a generalized linear model using R software. This model was $Y = \mathbf{1}\mu + \textbf{genotype} + \textbf{\textit{e}}$, where $\textbf{\textit{Y}}$ is the response vector of the LMD, $\mu$ is the mean of LMD, $\textbf{genotype}$ contains three different levels of genotypes (AA, AB, and BB). The $\textbf{genotype}$ was considered a fixed effect in the model, and $\textbf{\textit{e}}$ represents a vector of residual errors. For LMD, the least square means of different genotypes (AA, AB, and BB) were compared through Least Significant Difference (LSD) in R. An SNP with a p-value< 0.05 was considered to be significantly associated with LMD.

## Dual-luciferase expression assays of promoter and enhancer regions

For the SNPs that were identified to be significantly associated with LMD through the aforementioned steps, JASPAR [36] was used to identify transcription factor binding sites. For the five SNPs found in the enhancer region of *BMP2*, which impact transcription factor binding sites (S1 Fig), we conducted dual-luciferase expression assays to verify their functions. In the enhancer region, the 400-bp genome region flanking the significantly associated SNPs was cloned. As the distance between the SNP rs328487632 and rs319025934 is 24bp, we only cloned a single 800bp genome region containing these two SNPs. In total, four fragments in the enhancer region were cloned. Dual-luciferase expression experiments were performed for the promoter region of candidate genes, where the region (2.5 kb upstream) surrounding the transcription start site (TSS) was considered the promoter of the candidate genes.

Genomic DNA was extracted from the ears of Yorkshire pigs. The promoter and enhancer regions of candidate genes were amplified from genomic DNA through PCR using Phanta Max Super-Fidelity DNA Polymerase (P505, Vazyme). The PCR product was subsequently cloned into the pGL3- basic (E1751, Promega) and pGL3-promoter vectors (E1761, Promega) upstream of the luciferase gene for promoter and enhancer assays, respectively, using restriction enzymes NcoI (FD0574, Thermo) and KpnI (FD0524, Thermo). Dual-luciferase assays were conducted in PK15 (Porcine Kidney 15) cells which were cultured in a 37˚C incubator with 5% $CO_2$. PK15 cells were plated into 48-well plates and co-transfected with reporter vectors and pRL-TK renilla luciferase control vector (E2241, Promega) using Lipofectamine 2000 (11668500, Invitrogen). The transfected cells were incubated for 24 hours prior to lysis, which were lysed by using the Dual-Luciferase Reporter Assay System (11402ES60, Yeasen). For each construct, the reporter assay was repeated at least three times independently, and the results from a single representative experiment are presented in this study. Statistical analysis was conducted using a Student's t-test, and a p-value lower than 0.05 was considered significant.

### Electrophoretic mobility shift assays (EMSA)

PK15 nuclear extracts were prepared using NE-PER Nuclear and Cytoplasmic Extraction Reagents (78833, Thermo Fisher Scientific) and quantified using the BCA method (P0010S, Beyotime). Single-stranded and reverse-complement DNA probes were synthesized with or without a 5′-end biotin label. For the candidate SNPs, probes were designed as follows: for SNP rs1111440035 (M4 in the promoter, chr 17:15749990 bp), the M4-L-CC probe forward strand was 5′-CCCACCCGAACGACCTCGGGGCGA-3′; M4-H-GG probe forward strand was 5′-CCCACCCGAAGGACCTCGGGGCGA-3′; for SNP rs80791204 (M5 in the promoter, chr 17:15750750 bp), the M5-H-GG probe forward strand was 5′-AGGGAGAATAACTTGGGCT CCTCACTTCGCG-3′; and the M5-L-CC probe forward strand was 5′-AGGGAGAATAACT TGCGCTCCTCACTTCGCG-3′; for SNP rs321846600 (in the enhancer), the TT probe forward strand was 5′-GACAACCAGATCCATCTGGGCACCAGTC-3′; and the CC probe forward strand was 5′-GACAACCAGATCCACCTGGGCACCAGTC-3′.

An EMSA assay was performed using a Light Shift Chemiluminescent EMSA Kit (89880, 20148E, Thermo Fisher Scientific) according to the product manuals. We prepared and performed the binding reactions as follows: for the negative groups (lanes 1 and 4), 1 μL of 10× binding buffer, 0.5 μL of 1 μg/ μL poly(dI-dC), and 6.5 μL of $ddH_2O$; for the experiment groups (lanes 2 and 5), 1 μL (2 μg) of nuclear protein, 1 μL of 10× binding buffer, 0.5 μL of 1 μg/ μL poly(dI-dC), and 6.5 μL of $ddH_2O$; for the competition groups (lanes 3, 6, 7, and 8), 1 μL (2 μg) of nuclear protein, 1 μL of 10× binding buffer, 0.5 μL of 1 μg/ μL poly(dI-dC), 2 μL of 1 pmol/ μL unbiotin-labeled probes, and 4.5 μL of $ddH_2O$. Binding reactions were incubated at room temperature for 20 minutes, followed by an additional incubation at room temperature for 30 minutes after the addition of biotin-labeled probes. Then, 2.5 μL of 5× Loading Buffer was added to each 10μL reaction. To run the 6% polyacrylamide gel, the voltage was set to 100 V, and samples were electrophoresed until the bromophenol blue dye migrated approximately 3/4 through the length of the gel in 0.5× TBE buffer. We transferred the bands at 380 mA (~100V) for 60 minutes. Finally, the biotin-labeled DNA was detected using chemiluminescence.

## Results

### Descriptive statistical analysis and genetic parameters

In this study, we obtained 16,533 recordings of LMD in Yorkshire pigs. Descriptive statistics of the LMD recordings are listed in Table 1. The mean of LMD was 6.19 cm with a standard deviation (SD) of 0.74 cm. The additive genetic variance of LMD was 0.095, with a standard error

**Table 1. Descriptive statistics of loin muscle depth in the Yorkshire population.**

| Traits | #indiv | Min | Mean | Max | SD |
|---|---|---|---|---|---|
| Loin muscle depth (cm) | 16533 | 1.52 | 6.19 | 8.94 | 0.74 |

of 0.016. The residual variance was 0.160, with a standard error of 0.035. The estimated heritability of LMD was 0.373, with a standard error of 0.018.

## Five significant SNPs associated with LMD were identified by GWAS

A linear mixed model analysis was applied to perform the GWAS analysis of the LMD trait. The significance threshold was calculated as the cut-off after the Bonferroni correction. In total, 5 SNPs reached the significance threshold of 5.93 (−log10(0.05/42772) = 5.93) (Fig 1). All significantly associated SNPs were located within the region of 15.51 to 16.31 Mb on SSC17 (with a span of 0.8 Mb). Detailed information regarding the SNPs and their nearest genes is shown in Table 2.

## LMD-related SNPs are located in the same TAD

Our previous study identified the boundaries of topologically associated domains (TADs) using Hi-C data from pig muscle tissues. These boundaries served to limit chromatin

a

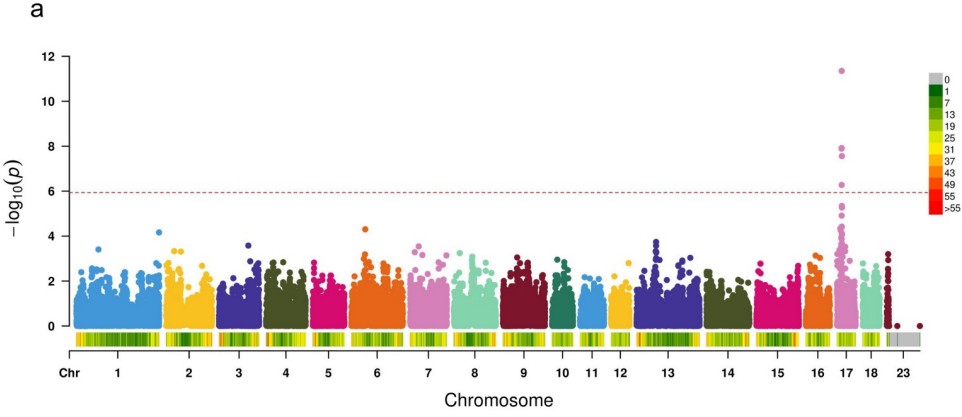

b

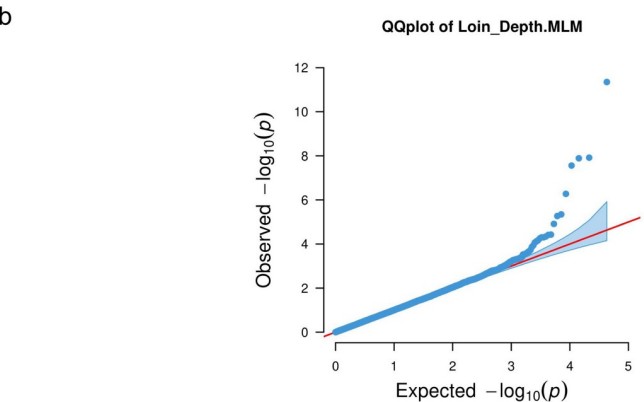

**Fig 1.** (a) Manhattan plot and (b) QQ plot showing genome-wide association for loin muscle depth (LMD) in Yorkshire pig. The red dotted line in (a) represents -log₁₀(p)-value = 5.9.

**Table 2. Summary of significantly associated SNPs for loin muscle depth (LMD) and relevant genes in Yorkshire pigs.**

| SNP | SSC | Position(bp) | Alleles | P value | Region | Nearest Gene |
|-----|-----|-------------|---------|---------|--------|-------------|
| WU_10.2_17_16896163 | 17 | 15518001 | C/T | 1.29E-08 | Intergenic | ENSSSCG00000043546 |
| WU_10.2_17_16951872 | 17 | 15534531 | G/A | 1.21E-08 | Intergenic | ENSSSCG00000043546 |
| WU_10.2_17_17013787 | 17 | 15710331 | C/T | 5.31E-07 | Intergenic | BMP2 |
| MARC0112426 | 17 | 15755711 | C/A | 4.47E-12 | intronic | BMP2 |
| WU_10.2_17_18106530 | 17 | 16312580 | T/C | 2.76E-08 | Intergenic | ENSSSCG00000025527 |

interactions, mediated by specific proteins, between different TADs in nuclei. The interaction effects of *cis*-regulatory elements within the same TADs were significantly stronger than those spanning the two nearest adjacent TADs [18]. Based on this finding, we integrated the GWAS results and Hi-C data to identify candidate interaction regions surrounding the SNPs that were significantly associated with LMD. Results suggested that these significantly associated SNPs (spanning 0.8 Mb and ranging from 15.51 Mb to 16.31 Mb on SSC17) were all located in the same TAD region (SSC17: 15.08 to 16.76 Mb). Three sub-domains covered all the significantly LMD-associated SNPs. Among these three sub-domains, one sub-domain (15.65 to 15.89 Mb) was completely within the 0.8-Mb significantly LMD-associated SNP region (15.51 to 16.31 Mb), while the other two were partially within it (Fig 2B). These results demonstrated that the significant SNP-located region had limited opportunity to interact with genome regions located in other TADs due to the presence of boundaries between different TADs.

We further investigated genes located in the aforementioned TAD region (15.08–16.76 Mb) on pig chromosome 17 and uncovered 11 genes located in this region. Of these 11 genes, seven were long non-coding RNA genes, whereas the other four genes were coding genes. Detailed information is shown in Fig 2 and S1 Table. This TAD region was investigated in the pigQTLdb database, showing that QTLs in this genomic region (15.08–16.76 Mb on SSC17) were associated with body weight, body height, intramuscular fat content, average daily gain, average backfat thickness, carcass length, and other traits (S2 Table), which may influence the loin muscle depth.

## Linkage disequilibrium analysis highlights the candidate QTL region harboring *BMP2*

To further characterize the candidate QTL region within the TAD (SSC17: 15080000–16760000), we investigated linkage disequilibrium (LD) patterns around the significantly LMD-associated SNPs. Four LD blocks were detected in this region using the confidence interval algorithm in Haploview software (Fig 3A). MARC0112426, the most significantly associated SNP, was located in the intron of *BMP2* (SSC17: 15750487–15762982) within LD block 2 (15.65 Mb to 15.75 Mb). Two significantly associated SNPs (WU_10.2_17_17013787 and MARC0112426) were completely linked (D' = 1) in LD block 2. We further studied the Hi-C contact map surrounding LD block 2. There was a complete sub-domain (15.65 to 15.89 Mb) covering LD block 2 (15.65 to 15.75 Mb) (Fig 3B), which indicated LD block 2 has a high opportunity to interact with genomic regions within this sub-domain. Although the SNPs located inside and outside of LD block 2 showed a high linkage disequilibrium (LD) with each other, the sub-domains isolate this disequilibrium. Thus, SNPs outside the sub-domains were considered to be irrelevant to the LMD trait. Overall, these results indicated that LD block 2 is a compelling candidate QTL region for the LMD trait.

Furthermore, LDLA results demonstrated that the block 2 region (15.65 to 15.75 Mb) was significantly associated with LMD (Fig 3C and S3 Table). The SNPs with the top six LRT values

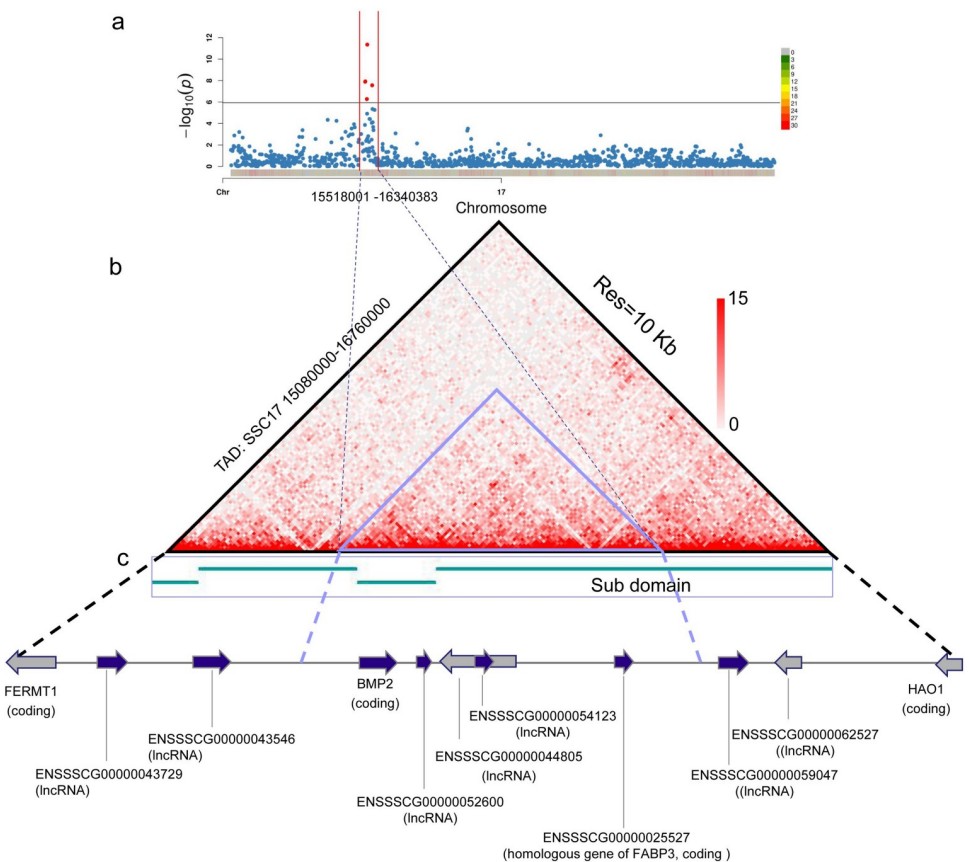

**Fig 2. Association results for SSC17.** (a) Manhattan plot of genome-wide p-values for pig chromosome 17. (b) Hi-C contact heatmap surrounding a significantly associated QTL region at 10 Kb resolution. (c) Map of characterized genes from 15.51 to 16.31 Mb on SSC17.

were within the 95% confidence interval of the highest LRT values [31,32], and the region (15.65 to 15.89 Mb) containing these six SNPs completely covered the identified LD block2 region (15.65 to 15.75 Mb). The SNP with the highest LRT value (WU_10.2_17_17075196) was located at SSC17:15689085. These results confirmed that the LD block2 region (SSC17:15659761–15755711) was an important candidate QTL region associated with LMD, aligned with the GWAS and Hi-C integrated results. Only one gene, *BMP2*, was located in this candidate QTL region, suggesting that *BMP2* is a candidate gene related to LMD (Fig 3C).

## SNPs in *cis*-regulatory elements of *BMP2* are significantly associated with LMD

*Cis*-regulatory elements of the pig genome were identified in our previous study [18]. To further comprehend the genetic mechanisms of LMD, we investigated the association between SNPs in the *cis*-regulatory elements of the pig genome and the LMD trait (Fig 4).

To further identify the functional variants, we analyzed SNPs using target region sequencing technology. For the 732 randomly selected individuals, 4,980 SNPs were captured in the GWAS candidate region (15.51 Mb to 16.31 Mb on SSC17). According to the association analysis, there were 184 SNPs significantly associated with LMD (Fig 4D and S4 Table), of which five SNPs, chr17:15674366 (rs321846600), chr17:15683553 (rs328487632), chr17:15683577 (rs319025934), chr17:15684170, and chr17:15724776 (rs321766789), were located in *cis*-

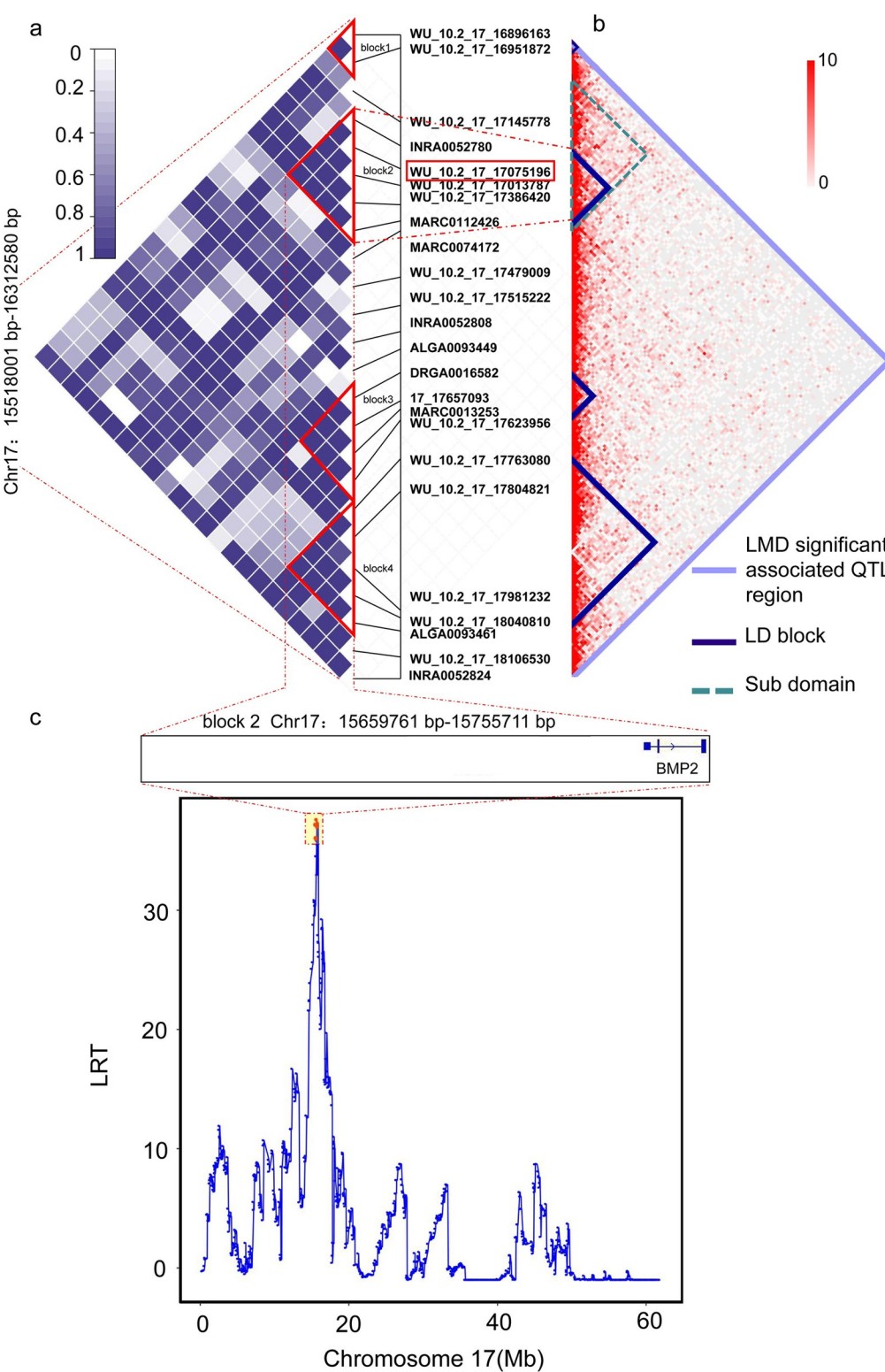

**Fig 3. Linkage analysis (LDLA) integrating Hi-C data locating target regions significantly associated with LMD.** (a) Linkage disequilibrium blocks were detected in the region (15.51 to 16.31 Mb on SSC17). SNPs in red boxes have the highest P value. As the chip is named in accordance with version 10.2, the figure uses version 10.2. The location has been converted to version 11.1 in subsequent analyses. (b) Hi-C contact heatmap of significant SNP locations (LMD significantly associated QTL region 15.51–16.31 Mb). (c) Likelihood ratio test (LRT) profiling using the combined

linkage disequilibrium and linkage analysis (LDLA) approach for LMD. The x-axis represents the scan steps of the analysis, and the y-axis represents the LRT value. The top six LRT values are denoted in red, and the 95% confidence interval (CI) defined by the LRT-dropoff-2 method is indicated in a red-shaded block.

regulatory elements of the pig genome. These five SNPs were distributed in the enhancer region of *BMP2*. Moreover, the Hi-C contact map and enhancer-gene correlation results suggested that enhancers in the LMD QTL regions interacted with the *BMP2* promoter (Fig 4A and 4B). These results suggested that the identified SNPs in the *cis*-regulatory elements of *BMP2* are associated with LMD in pigs.

## Important candidate variant scanning in the enhancer region of *BMP2*

Based on the target region sequencing results, five significantly associated SNPs were located in the *BMP2* enhancer region. SNPs in regulatory regions may function by modulating transcription factor binding. Therefore, motif analyses were performed to further validate the functional consequences of these five SNPs. Results suggested that these five SNPs may all disturb the TF binding motif (S1 Fig).

We hypothesized that the genomic regions harboring these five SNPs are able to regulate the LMD by altering enhancer function. To validate our hypothesis, the 400-bp genomic region flanking each significantly associated SNPs with different allelic combinations was cloned into pGL3-promoter luciferase reporter vectors respectively. Because the distance between SNP rs328487632 and rs319025934 is only 24 bp, we cloned one 800-bp genome region containing these two SNPs. From reporter assays, we observed stronger luciferase activity for each cloned genomic region compared to the empty vector, supporting the hypothesis that these five SNPs can function as enhancers (Fig 5A–5D). Furthermore, for the cloned 800-bp genome region surrounding the five aforementioned SNPs, we compared the luciferase activities between alleles. Results showed that the enhancer activity for SNPs rs328487632&rs319025934, and rs321766789 did not show a significant change between alleles (Fig 5A and 5B). The cloned 800-bp genomic region covering SNP 15684170 contains more than one SNP in the region. To exclude effects on enhancer activity from other SNPs, in addition to comparing the enhancer activity between the T and G substitution at position 15684170, we mutated the T to a G in position 15684170. Results (Fig 5D) showed that the enhancer activity did not change significantly, indicating that SNP 15684170 is not an important site for enhancer activity.

Among these five SNPs, the presence of a C at rs321846600 showed significantly higher enhancer activity than a T (Fig 5C). This implied that the SNP rs321846600 in the enhancer region of *BMP2* may be important for the regulation of *BMP2* expression. Thus, we considered the SNP rs321846600 as a candidate variant that may be functionally related to the LMD trait.

## Variant scanning in the promoter of *BMP2* identified two functional mutations

We further characterized the *BMP2* promoter and investigated its polymorphisms. We cloned the promoter region of porcine *BMP2* from 10 Yorkshire pigs to scan for potential variants, and five variants (4 SNPs and one Indel) were identified. These five variants are completely linked and formed two haplotypes: T(CAAAC)TGG (BMP2 H) and C(T—-)ACC (BMP2 L) (Fig 6A). The promoter with the haplotype T(CAAAC)TGG (BMP2 H) had higher transcription activity than C(T—-)ACC (BMP2 L), as found by the dual-luciferase reporter assays (Fig 6B).

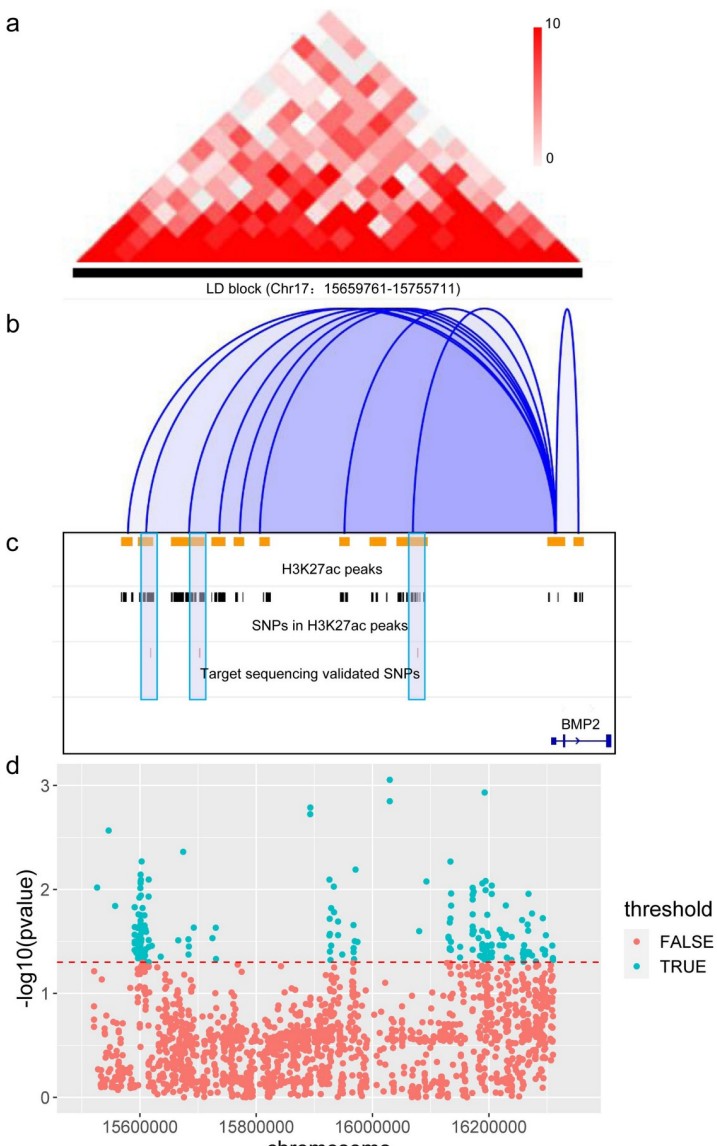

**Fig 4. Visualization of SNPs using different methods or validation locating in significant H3K27ac ChIP-seq peak regions from various tissues [18].** (a) Hi-C contact heatmap in the candidate QTL region (LD block 2, chr 17: 15659761–1575571) of LMD. (b) Significant enhancer-gene pairs in the candidate QTL region of LMD [18]. (c) SNPs from the Yorkshire pig whole genome sequence data, and target sequencing validated in H3K27ac ChIP-seq peak regions of various tissues. The purple box and sky blue box represent the significant (P<0.05) SNPs validated by target sequencing. (d) Scatter plot showing single-SNP trait association analysis of targeted sequencing data. The red dotted line in (d) represents a p-value = 0.05.

To identify important candidate variants influencing promoter activities between BMP2 H and BMP2 L, we created mutant reporter vectors (BMP2 H>L and BMP2 L>H) based on the five identified SNPs (Fig 6C). The reporter assay suggested that the promoter activity had changed significantly (Fig 6D), which suggested that M4 (rs1111440035, chr 17:15749990 bp) or M5 (rs80791204, chr 17:15750750 bp) altered the promoter activities of *BMP2*. The original alleles in the M4 (BMP2 M4 H>L) and M5 (BMP2 M5 H>L) of BMP2 H were mutated to the corresponding alternative alleles one nucleotide at a time (Fig 6E). Results showed that when

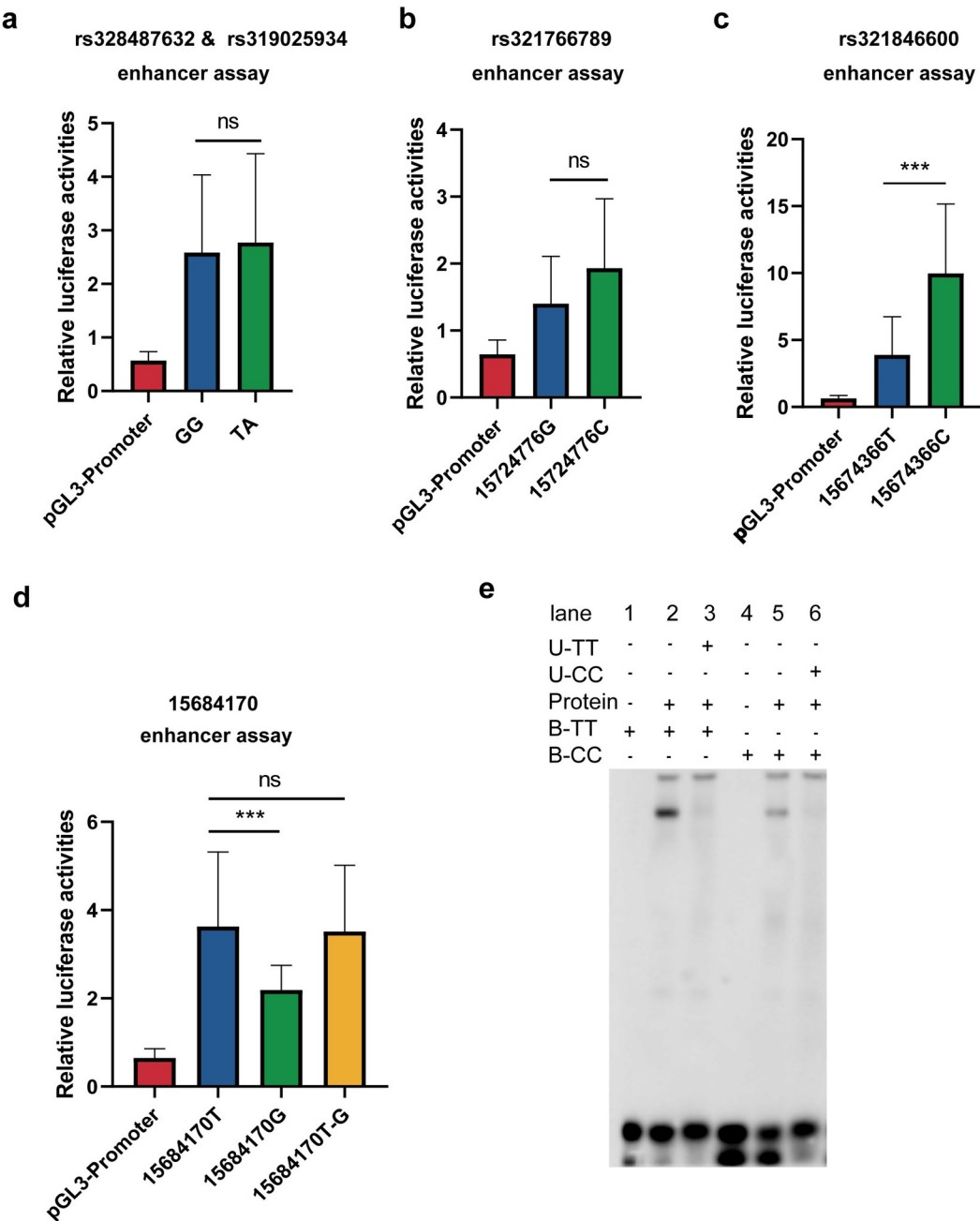

**Fig 5. Dual-luciferase reporter and EMSA assays.** (a–d) Luciferase reporter assays containing the SNPs 15683553 (rs328487632), 15683577(rs319025934), 15724776(rs321766789), 15684170, and 15674366(rs32184660), for the enhancer assay. (d) 15684170T-G represents the SNP 15684170 mutating a T into a G. Expression levels were measured after transfection into the porcine kidney cell line, PK15. The pGL3 promoter vector containing the SV40 promoter was used for the enhancer assay. Luciferase signals were normalized to Renilla signals (n = 3). Data are presented as mean ± SEM, and $p$ values were calculated using a Student's $t$-test. * $p \leq 0.05$; ** $p \leq 0.01$; *** $p \leq 0.001$. (e) EMSA results of rs321846600. U-TT represents the unlabeled TT probe, similar to U-CC, and B-TT represents the biotin-labeled TT probe, similar to B-CC. Three replicates were performed for each experiment.

the alleles in either the M4 (G>C) or M5 (G>C) sites of BMP2 H were mutated, the promoter activity significantly decreased (Fig 6F). Therefore, we concluded that both rs1111440035 and rs80791204 caused different promoter activities of BMP2 H and BMP2 L.

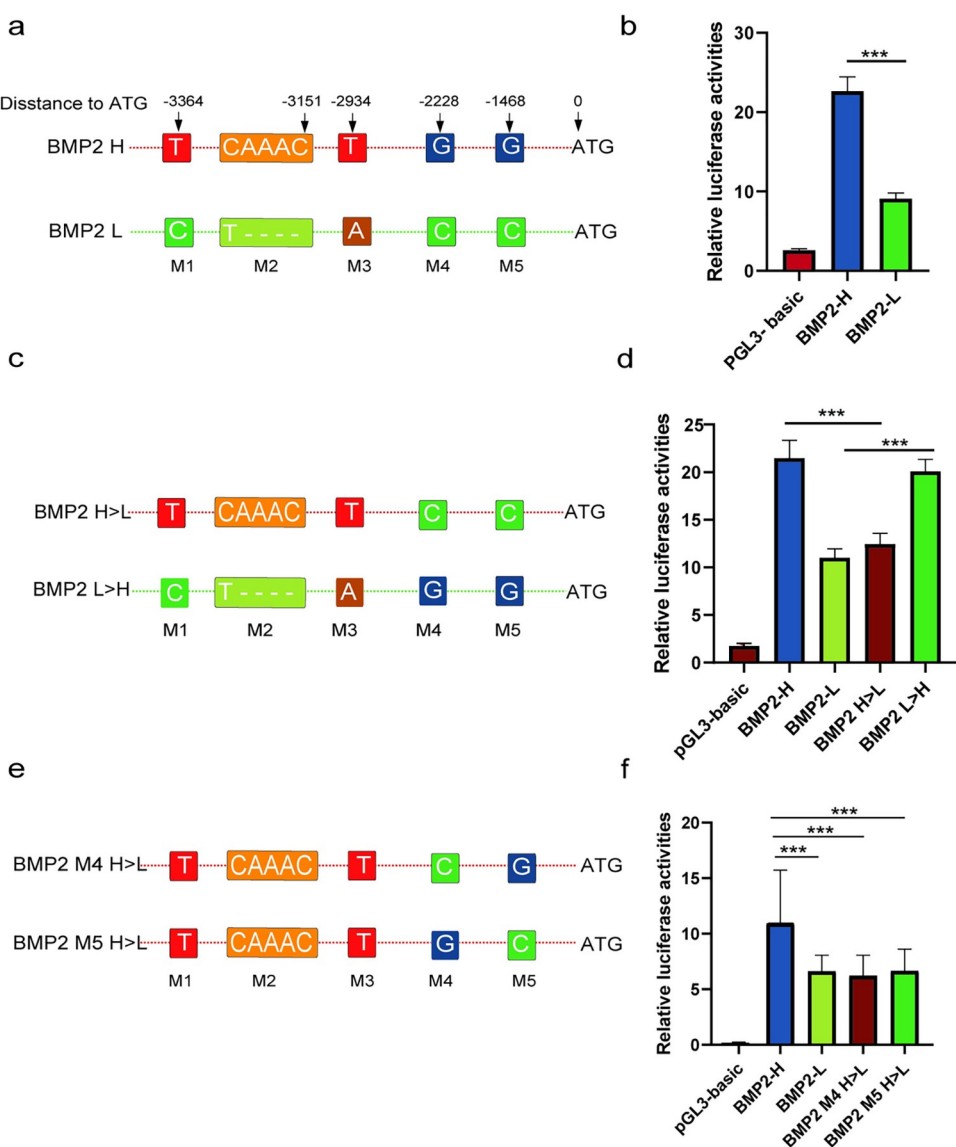

**Fig 6. Identification of functional mutations in the porcine *BMP2* promoter.** Luciferase reporter assays using vectors containing the SNP locus for the promoter assay. (a)(c)(e) SNP location of each vector. The transcriptional start site (TSS) was defined as +1. M1 represents the vector containing mutations in the first sites. (b)(d)(f) Luciferase reporter assays using vectors containing the SNP locus for promoter assay. The pGL3 basic vector lacking a promoter was used for the promoter assay. Luciferase signals were normalized to Renilla signals. Each assay involved three independent experiments. The results from one representative experiment in the porcine kidney cell line, PK15, are shown. Data are presented as mean ± SEM, and *p* values were calculated using a Student's *t*-test. $^{*}$ $p \leq 0.05$; $^{**}$ $p \leq 0.01$; $^{***}$ $p \leq 0.001$.

To further validate the identified candidate SNPs, motif analysis was conducted for the two SNPs. Motif analysis showed that rs1111440035 (M4, C>G) disturbed the binding of the transcription factor NR2C1 (Fig 7A). Similarly, rs80791204 (M5, G>C) overlaps with a predicted HLTF binding motif (Fig 7B). Moreover, multiple sequence alignments (Fig 7C and 7D) revealed that the rs1111440035(M4) within the porcine *BMP2* promoter was conserved across mammals. Genomic Evolutionary Rate Profiling (GERP) scores are often used to measure the conserved nature of gene sequences across species, and a high GERP score suggests that sequence is highly conserved. We added GERP scores to determine the conservation of M4

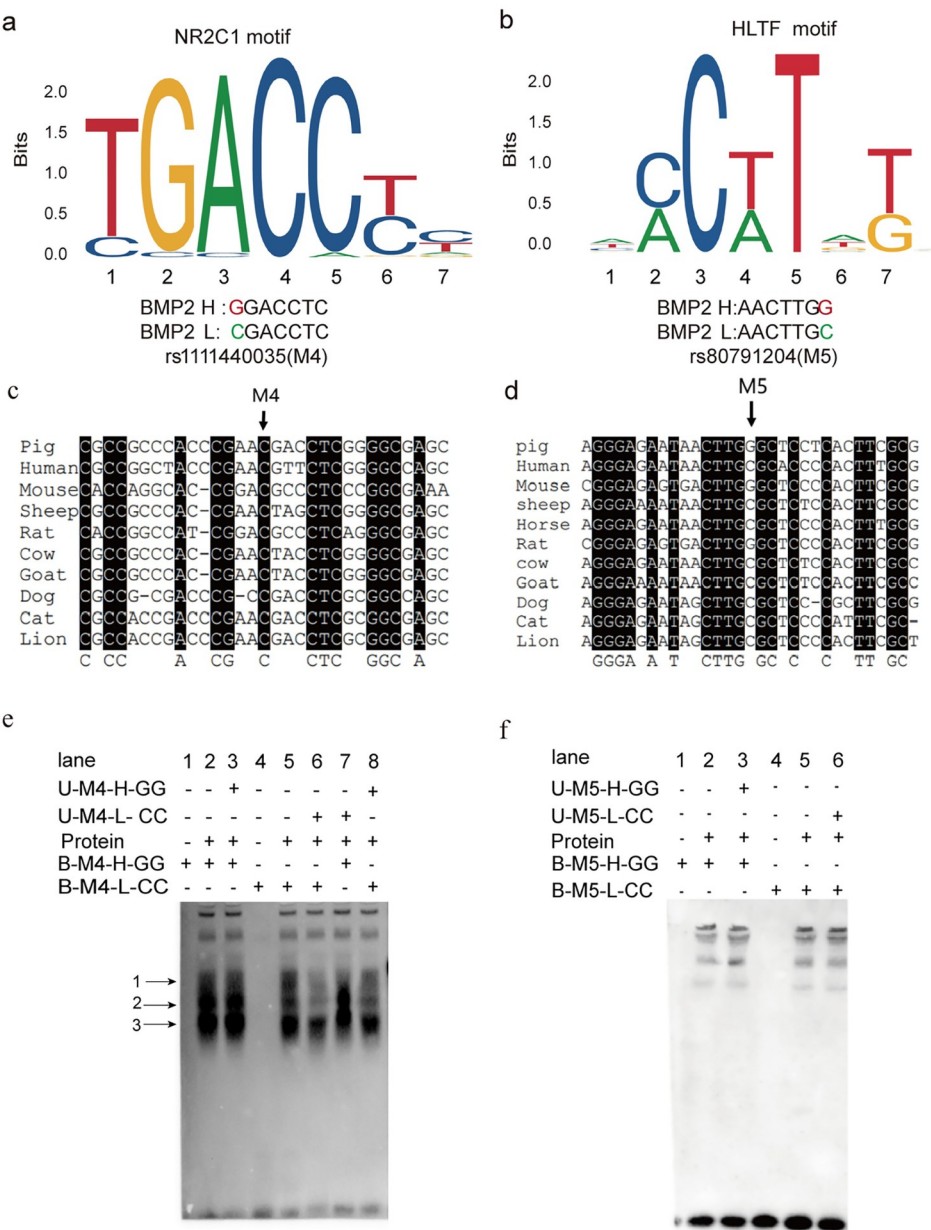

**Fig 7. Analysis and EMSA assay for rs1111440035 and rs80791204.** (a)(b) Motif analysis for rs1111440035(M4) and rs80791204(M5). (c)(d) Multiple sequence alignment across mammalian DNA elements surrounding the M4 and M5 mutation sites. (e)(f) The EMSA results. Here, "protein" represents PK15 nuclear extracts. U-M4-L-CC represents unlabeled M4-L-CC probe, similar to U-M4-H-GG, U-M5-L-CC, and U-M5-H-GG, while B-M4-L-CC represents biotin-labeled M4-L-CC probe, similar to B-M4-H-GG, B-M5-L-CC, and B-M5-H-GG. Three replicates were performed for each experiment.

and M5. For M4, the GERP score is 1.91; for M5, the GERP score is −5.13. These GERP scores shows that M5 is less conserved than M4.

## Verification of important candidate variants by EMSA

Next, we performed EMSAs using nuclear proteins from PK15 cells to evaluate binding affinities for important candidate variants (SNP rs321846600 in the enhancer region; rs1111440035

and rs80791204 in the promoter region). Results showed that, for rs321846600, a T>C mutation reduced the DNA–protein binding affinity (Fig 5E). For rs1111440035, both the M4-L-CC and M4-H-GG probes detected two common bands (Fig 7E, arrows 2 and 3 in lanes 2 and 5), whereas the M4-L-CC probe detected an additional shifted band (Fig 7E, arrow 1 in lane 5). When using U- M4-H-GG to compete with B- M4-L-CC, the shift was not eliminated (Fig 7E, arrow 1 in lane 8). Thus, the specific band suggested that unknown *trans*-acting factors could uniquely bind to the M4-L-CC probe and likely respond to the different promoter activities of BMP2 H and BMP2 L. However, for the M5-L-CC and M5-H-GG probes for rs80791204, no additional shifted bands were detected (Fig 7F), suggesting that these alleles are not different in binding affinity. Combined with the above results, this suggests that SNPs rs321846600 and rs1111440035 influence transcription factor binding and may alter the expression of the *BMP2* gene, which thus impacts LMD trait expression.

## Discussion

In this study, we pre-corrected LMD phenotypes and genotyped the DNA of 2,733 Yorkshire pigs using a 50k SNP chip for GWAS. Five SNPs in the genomic region between 15.51 and 16.31 Mb on SSC17 were significantly associated with LMD. To fully understand the GWAS results, we combined GWAS, LD block analysis, and Hi-C data to map the candidate genomic region related to LMD. Compared to traditional linkage analysis, LDLA is able to improve QTL detection and accurately map QTL locations [37,38]. Combined with the LDLA results, a ~10 kb QTL region (LD block2, 15.65 Mb to 15.75 Mb) and the *BMP2* gene on chromosome 17 were associated with LMD in Yorkshire pigs. To further detect important functional variants in the transcriptional regulatory region, we used target region sequencing technology to increase the SNP density of the candidate genomic region. Together with SNP association analysis, obtaining information on *cis*-regulatory elements, and motif analysis, we were able to identify five variants were as important functional variant candidates. Dual-luciferase expression assays and EMSAs were used to evaluate these important candidate variants. Correspondingly, one SNP was detected that impacts enhancer activity and one SNP was detected that affects promoter activity of the *BMP2* gene. Compared to traditional research methods which only used GWAS or integrated GWAS with gene expression data [39–41], our method provides conclusive evidence of a major QTL affecting LMD on chromosome 17 and further identifies candidate SNPs with functional relationships to LMD.

Genome-wide association studies have been extensively used to identify genomic regions associated with important economic traits, however, revealing trait-related genetic mechanisms is still a research challenge. To identify the important candidate variants, 3D epigenomic characteristics were used to uncover important candidate functional sites as follows: 1) The TAD information surrounding the LMD-associated SNPs was used to dissect the SNP functions, which showed the genome region of significantly associated SNPs has limited opportunity to interact with genomic regions in other TADs. 2) The identified candidate genomic region (15.51 to 16.31 Mb) fully covers one sub-domain, and LD block 2 is located within this sub-domain. According to the property of sub-domains, the SNPs within LD block 2 have higher probability of interaction with genomic regions within the sub-domain. We admitted that we cannot totally exclude the possibility that variants outside block 2 may contribute to the LMD QTL effects, but the possibilities are much lower than the block 2. 3) The epigenetic data (enhancers and promoters) help to further identify candidate SNPs based on target re-sequencing data. 4) The dual-luciferase expression assays and EMSAs assist in validating the identified important variants. Overall, integrated analysis of genome-wide association studies and 3D epigenomic characteristics aid in understanding the function of variants governing LMD.

Loin muscle depth is an important economic trait that influences the lean meat content of pigs. Revealing the genetic mechanism and identifying important markers contribute to the genetic improvement of LMD. LD analysis suggested that there were four blocks in the LMD significantly associated region, and these four blocks were found in the LMD-associated region and are related to body weight, body height, intramuscular fat content, average daily gain, backfat, and carcass length. LMD has been reported to be negatively correlated with backfat and positively correlated with carcass length [42,43]. Additionally, the most significant SNP in this LMD-associated region has been reported to be associated with LMA [44]. Therefore, the LMD-associated region may be the LMD QTL region.

In our study, *BMP2* was identified to be the candidate gene for LMD. *BMP2*, a member of the superfamily transforming growth factor beta (TGF-beta), has been reported to participate in adipogenesis [45–47], myogenesis [48,49], chondrogenesis, and osteogenesis [50–52]. *BMPs*, which are also known as activators or inhibitors, participate in specific stages of muscle progenitor cell differentiation during skeletal muscle development and regeneration [53–59]. Trait association analysis of human skeletal muscle volume revealed that *BMP2* is associated with increased skeletal muscle volume [60]. In previous studies, *BMP2* has been found to be related to carcass length [61,62], loin muscle area, body size, and several foot and leg (FL) structural soundness traits in pigs [44]. Endogenously or exogenously expressed *BMP2* promotes adipogenesis [45,63]. Furthermore, *BMP2* has been found to induce the upregulation of adipogenic gene expression, leading to increased intramuscular fat (IMF) deposition in castrated animals [64]. It has also been reported that LMD and LMA are negatively correlated with backfat depth and positively correlated with carcass length [42,43]. This evidence supports the hypothesis that *BMP2* is a functional candidate gene regulating LMD in Yorkshire pigs.

Overall, our work suggested that *BMP2* expression plays a crucial role in driving LMD variation in Yorkshire pigs and that the C allele of rs321846600 reduces DNA–protein binding ability, which increased BMP2 enhancer activity. We identified two mutation sites in the promoter region of porcine *BMP2*, forming two haplotypes. The G alleles of rs1111440035 and rs80791204 are favorable as they can increase *BMP2* promoter activity in cell reporter assays. Moreover, EMSA results indicated that the C allele at rs1111440035 binds to an unknown transcription factor, which leads to the C allele at rs1111440035 displaying weaker promoter activity than the G allele. To our knowledge, no transcriptional factors directly regulating *BMP2* expression have previously been reported. In the future, we will perform supershift assays and protein pull-down experiments to characterize the transcription factors for SNP rs1111440035 (C>G) and rs321846600 (T>C) and explore the mechanism of regulation of *BMP2* expression. Overall, our results indicate that both rs1111440035 and rs321846600 are likely important candidate mutations impacting LMD in Yorkshire pigs.

## Conclusion

In our study, GWAS and epigenomics data, including ChIP-seq data and Hi-C data, were integrated to identify an LMD-related QTL region, as well as candidate genes within this region. A ~10 kb region (15659761 bp to 15755711 bp) on SSC17 was identified and determined to be the candidate LMD QTL region. The *BMP2* gene was identified as the candidate gene most likely to be associated with LMD. SNPs rs1111440035 and rs321846600 are likely important candidate mutations impacting LMD in Yorkshire pigs. Our study made the first attempt to identify the major candidate genetic variants and candidate genes regulating an important production trait (LMD) in pigs through the integration of genome-wide association studies and 3D epigenomics.

## Supporting information

**S1 Table. Description of gene information in the QTL region significantly associated with LMD.**
(XLS)

**S2 Table. Description of quantitative traits loci (QTL) in the regions significantly associated with LMD.**
(XLS)

**S3 Table. Summary of linkage disequilibrium and linkage analysis (LDLA).**
(XLS)

**S4 Table. Description of information for captured sequencing loci significantly associated with LMD.**
(XLS)

**S1 Fig. Motif analysis for the SNPs significantly associated with LMD by target regional sequencing.**
(TIF)

**S1 Data. The data of phenotype.**
(ZIP)

**S2 Data. The data of genotype (This file can be opened with Notepad++).**
(ZIP)

## Author Contributions

**Conceptualization:** Yuanxin Miao, Yunxia Zhao, Tao Xiang.

**Formal analysis:** Yuanxin Miao, Yunxia Zhao, Siqi Wan.

**Funding acquisition:** Yuanxin Miao, Shuhong Zhao, Tao Xiang.

**Methodology:** Yunxia Zhao, Xuewen Xu, Tao Xiang.

**Software:** Yuanxin Miao, Siqi Wan, Quanshun Mei, Chuanke Fu.

**Writing – original draft:** Yuanxin Miao, Yunxia Zhao, Siqi Wan, Tao Xiang.

**Writing – review & editing:** Heng Wang, Xinyun Li, Shuhong Zhao, Xuewen Xu.

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
