## [Decision Letter · Decision Letter 0]

11 Jan 2023

Dear Dr Xiang,

Thank you very much for submitting your Research Article entitled 'Integrated analysis of genome-wide association studies and 3D epigenomic characteristics reveal the BMP2 gene regulating loin muscle depth in Yorkshire pigs' to PLOS Genetics.

The manuscript was fully evaluated at the editorial level and by independent peer reviewers. The reviewers appreciated the attention to an important problem, but raised some substantial concerns about the current manuscript. Based on the reviews, we will not be able to accept this version of the manuscript, but we would be willing to review a much-revised version. We cannot, of course, promise publication at that time.

If you decide to revise the manuscript for further consideration at PLOS Genetics, please aim to resubmit within the next 60 days, unless it will take extra time to address the concerns of the reviewers, in which case we would appreciate an expected resubmission date by email to plosgenetics@plos.org.

We are sorry that we cannot be more positive about your manuscript at this stage. Please do not hesitate to contact us if you have any concerns or questions.

Yours sincerely,

Martien Groenen, PhD

Academic Editor

PLOS Genetics

Gregory Barsh

Editor-in-Chief

PLOS Genetics

While the manuscript provides clear evidence for a QTL affecting LMD, the results do not support the claim that two SNPs upstream of BMP2 are causative. Furthermore, I agree with reviewers 1 and 3 that the epigenetic data cannot be used to identify the causative variants within the region identified and therefore cannot be presented as a method for fine mapping QTLs. Both claims need to be toned down in the revised version of the manuscript

Reviewer's Responses to Questions

**Comments to the Authors:**

Reviewer #1: This paper provides conclusive evidence for a major QTL affecting loin muscle depth (LMD) on chromosome 17 in a population of Yorkshire pigs and suggestive evidence for the functional significance for some SNPs located in putative regulatory regions of BMP2. The problem is that it is unknown if the candidate SNPs affect LMD or not. Therefore, the claim that two SNPs upstream of BMP2 are causative (L 49) is not justified. The authors do not have the genetic resolution to identify causal variants due to the broad QTL region (800 kb) with high LD among markers (Fig. 3). The initial mapping was done with a sparse SNP set, and the resequencing analysis carried out in this analysis show that the most strongly associated markers is located more than 200 kb away from the SNPs now claimed to be causative. The data are worth publishing but the authors need to tone down the arguments that these SNPs are causal it is more correct to state that these are candidate SNPs that may be functionally related to the LMD QTL. Also, I disagree with the statement that this paper presents a method for fine mapping QTLs. The epigenetic data used here are instructive for identifying functionally important regions of the genome but they cannot be used to exclude sequence variants from being causative and that is what is needed to fine map a QTL. This is the strength of genetic methods like linkage and LD mapping namely that they can exclude regions and thereby narrow down regions associated with a phenotype. What can be done as regards this locus would be to collect similar data from another pig population and also to search for sweep signals in this region. If this is a major locus under strong selection a clear sweep signal is expected among pigs selected for lean meat content. An alternative explanation is that this QTL is due to a haplotype effect caused by multiple closely linked QTLs within this interval so the effect of this region varies among pig populations.

Specific comments

Fig. 3. Please indicate coordinates for the region covered in Fig. 3A in order for readers to compare the regions mentioned in the text. Furthermore, clarify that all SNP data here are based on the sparse SNP chip. This figure illustrates the problem with the identification of causal variants because there is high LD across a large genomic region. The authors focus on the block2 region where BMP2 is located but SNPs in this region show complete or near complete LD with SNPs at the other end of the broad interval about 1 Mb away. Furthermore, it is confusing that the LRT peak is at around 9 Mb on chr 17 according to Fig. 3C whereas the text indicates that the peak is around 15.7 Mb??

L 298-305, resequencing analysis. The authors focus on the block 2 region 15.65-15.75 Mb, but this resequencing analysis (Table S4) shows that the most significantly associated SNPs are located in the interval 15.89-16.19Mb. The top SNP is located at 16.03 MB more than 280 kb from the candidate SNPs that are functionally evaluated. This is an important result that should be presented in a graph in one of the main figures. Since this is based on a dense SNP data set it provide information about the strength of association across the QTL region.

L 320. I assume you should cite Fig. 5 here although Fig 5 shows results for only 5 regions and not 6 as indicated in the text. Furthermore, it is not the SNPs that can function as an enhancer, it is regions harboring the SNPs that may have enhancer function. Three constructs are used in Fig 5d: T, G and T-G. The difference between G and T-G is not clear to me and is not explained in the legend. Also, why is this not a candidate SNP when G vs T is significantly different?

L 327-335. The description of the promoter polymorphism is a bit confusing. The authors refer to 5 SNPs but Fig. 6 suggests that the correct description is 4 SNPs + an InDel (5bp deletion or insertion), correct? Furthermore, the authors refer to rs numbers in the text but the figures use distance to TSS, these two annotations need to be connected in the text perhaps using the M1 to M5 annotations. Fig. 5e is confusing since constructs indicated below the bar is not annotated and do not seem to have been used so why include them in the figure??

Fig 7cd. There is much more comparative data available for mammals that can be used to assess conservation scores for these SNPs. The authors state that both M4 and M5 are evolutionary conserved but that is not correct for M5 because Fig. 7d shows that both C and G occur among other mammals. It would be better to refer these SNPs as REF and non-REF because these are polymorphism. For instance, for M4 the variant denoted MUT is the one present in all other mammals included here.

Gel shift assays, Fig 5f and Fig. 7ef. Firstly, it is not possible to verify causative SNPs with gel shifts, it can merely support their functional significance but it cannot prove causality, change subtitle. The authors should add one more lane to these experiments and test if excess cold probe of the other allele can compete for binding or not. If a mutation inactivates binding, cold mutation probe will not be able to compete with the labeled oligo binding the nuclear factor. For instance, Fig 5f shows that Cold U-WT can compete with B-WT but it is of interest to know if cold U-MUT can compete with B-WT or not. Fig. 7e has a lot of background and is not convincing, it needs to be improved. The authors highlight three bands labeled 1, 2 and 3, they should explain their interpretation which of these are specific and which are not.

L 367. This is not correct that you fine mapped the QTL you still have a large region, but you provided functional evaluation of candidate SNPs.

L. 388-389 I disagree with this statement, it is important to distinguish genetic evidence and functional evidence (see main comment above).

Minor comments

L 68-70. It is not the number of SNPs that is most relevant here but the number of independent loci associated with the trait, because a single QTL may involve hundreds or thousands of SNPs that show statistical significance if whole genome sequencing has been performed.

L 250. Us the same format consistently for genomic regions, so this should be indicated as 15.08 – 16.76 Mb

L 254 I think it is better to write that “the LMD-related causal variants were expected…”. This is because you use genetic data to map causal variants but if these are regulatory they may act on distance and the genes regulated by the causal variants maybe located outside the interval.

L. 265, change “mapping” to “map”, one example that some further improvement of English language is needed. Furthermore, should this be “QTLs” in pluralis? I assume you identified a single QTL that may include multiple causal variants but if so, these segregate as a single QTL. (However, the major QTL maybe composed of multiple closely linked QTLs but the authors do not have the genetic resolution to distinguish this architecture from a single QTL scenario).

L 285 Delete “The” at the beginning of the sentence, I assume you have not yet characterized all cis-regulatory elements in the porcine genome.

L 293 “two pig subpopulations” please clarify if these are selected from the same Yorkshire population used for the initial QTL mapping.

L301-305 This sentence is incomplete and the language must be corrected.

L 396, change to backfat

L 420-421 It is more appropriate to write binds “an unknown transcription factor”.

Table 2 is sorted by P-value but it is better to sort it based on genomic position since that gives a hint of the peak of association.

L 686, one decimal place is sufficient here and replace “p-value” with “-log10(p)-value)”

Fig. 2B. Why is this one single TAD-region, there is not much interaction between the 15.08 and 16.76 Mb

Reviewer #2: Cograts to the authors for the good piece of scientific work showing the have identified not only a major gene, but also a QTN. In my opinion, all the cahnges suggested by the reviewers were accomplished and Engish is in good shape

Reviewer #3: This paper describes a study in which a very strong statistical association was detected between a region on pig chromosome 17 and a key phenotype of commercial pigs, and subsequently, well-chosen genetic methods were used to dissect the functional differences between SNPs associated with the detected association.

General comments

Overall, this is a very interesting paper. The integration of functional analyses with the GWAS and LD-based analyses provide compelling results that implicate specific SNPs from the BMP2 gene in the association with loin muscle depth. My main concern, which could be addressed with rewording in various places, is that the title, abstract and conclusions state that 3D epigenomic characteristics helped them to dissect the basis of the genetic association but I do not see this. As I read the paper, the epigenomic characteristics did not really contribute to detecting the association or narrowing down the candidate SNPs, and rather, the GWAS and LD-based analyses were sufficient to detect the association. What I see as the novelty of the study is the effective use of the dual-luciferase and electrophoretic mobility shift assays to help identify which SNPs drive the strong association detected by GWAS/LDLA.

Another issue in the manuscript is that the results based on target region sequencing are not well integrated with those for the SNP array results. As I understand it, the target region sequencing work was added in response to reviewers’ comments on the previous version of the manuscript and these results make an important contribution to the paper. However, the two sets of results need to be combined better to provide a more logical explanation of the study design and results. This applies to the Methods (161-179), Results (285-308) and Discussion (371-389) (some specific issues are also detailed below in Additional comments).

There are some other sections where further details are required (see Additional comments, below). Finally, there are various grammar and wording problems. I have suggested some corrections, but further editing is required in some sections (see Grammar/wording, below).

Additional comments

Methods

107-135: I was confused by the treatment of phenotypes such that polygenic effects have been fitted twice (pre-correction + GWAS), using two different approaches. The authors should explain/justify this further.

153-160: The authors should provide further (brief) details here (the reader shouldn’t need to read the other paper to understand what has been done).

163-165: Seven out of how many SNPs?

164, 169, 170: Why 15.65-15.75 Mb on line 164, then subsequently 15.51-16.31 Mb?

167: Were the 732 individuals selected from high- and low-LMD groups or across the whole population?

167-169: Need to provide more details for the target region sequencing.

165, 172…: Which dataset was used for this association analysis described on 174-179? The 393 animals from high- and low-LMD or the 732 individuals?

180-202: I found this section confusing and further clarification is necessary. It was not clear to me which groups were being compared, i.e. What is meant by “experimental” and “control” groups (202)? Also, which pig(s) were used to generate the cells and provide DNA (189)?

207: As I understand, this was performed for 3 SNPs. Assuming so, this should be stated in the text (e.g. “For the three identified candidate SNPs…”)

Results

244-246: More details are required regarding the TADs. The reader should not need to read Reference [18] in order to understand this paper. Also see wording comment below.

284-308: As mentioned above, these results are not integrated clearly with the results from the KASP genotyping. For example, why wasn’t there any overlap between the two sets of identified SNPs? More generally, the results of the two approaches should be linked as they are addressing the same aim.

353-362: This paragraph needs more details on how to interpret these results, in particular for rs1111440035 or rs80791204. For example, “no additional shifted bands were detected” (359-360), should be following by “suggesting that these alleles did not differ in binding affinity” (if that is correct?). I could not understand the results for rs1111440035 (357-360).

Discussion

371-389: As mentioned above, this section needs editing to give a more logical explanation of the study design and interpretation of results.

Grammar/wording

76: change “variations” to “variation” (2 places); change “these” to “this”

102: reword, e.g. “SNPs were mapped to the Sscrofa11.1 pig genome assembly.”

108: reword, the ssGBLUP analysis incorporates both genetic and non-genetic effects.

150: add “A” before “likelihood ratio test”

170-171: needs rewording (I don’t understand what is meant here by “meanwhile distributed”)

173: change to “R software” or “R”

180: change to “…promoter and enhancer regions”

182: change “predict” to “identify”

183: change to “affect a transcription factor binding site”

185-188: This section needs rewording

--be more precise about their proximity of the two SNPs

--I don’t follow why 5 fragments were cloned

--be more precise about “Similar experiments…”

--change “executed” to “carried out”

190: should this be “from genomic DNA”?

192: change “assay” to “assays”

194: change “cell” to “cells”

217-229: this section requires editing to correct grammar (e.g. verb tenses) and wording

234-236: change “standard errors” to “standard error” (3 places)

238: change to “A linear mixed model analysis was applied…”

245: need to reword “insulation of interactions” (I don’t have any idea what this means)

250: change to “in the same TAD region”

256: add “the” before “above-mentioned”

265-2666: change “performed the linkage disequilibrium (LD) analysis” to “investigated linkage disequilibrium (LD) patterns”

274: change “vital” to another word (“compelling”?)

287-288: change to “Based on whole genome sequencing data for 60 Yorkshire pigs”

288: remove “the”

289: change to “the pig genome”

290: should mention this is on SSC17

298-308: this section requires editing to correct various grammar and wording problems

320: change to “supporting the hypothesis that all 6 of these SNPs can function as enhancers”

322: change to “in the enhancer region”

323: change “rest” to “remaining”

340: change “affecting” to “affect”

344: change “differential” to “different”

357: add “the” before “T>C”

359: change “probe” to “probes”

360-362: change to “…this suggests that SNPs rs321846600 and rs1111440035 influence transcription factor binding and affect…”

392: remove “the”

392-393: change to “four blocks are found in the LMB-associated region”

396: typo (“backfat”)

397: do you mean “the most significant SNP”?

398-399: reword sentence; does “this region” refer to that reported in Reference 44?

413: change to “This evidence supports the hypothesis that …”

415-427: this section requires editing to correct grammar and improve wording

**Have all data underlying the figures and results presented in the manuscript been provided?**

Reviewer #1: Yes

Reviewer #2: Yes

Reviewer #3: **No: **I presume that the journal will check that the data is made available if the paper is accepted. Currently the data is not available.

PLOS authors have the option to publish the peer review history of their article (what does this mean?). If published, this will include your full peer review and any attached files.

Reviewer #1: No

Reviewer #2: No

Reviewer #3: No

---

## [Decision Letter · Decision Letter 1]

7 May 2023

Dear Dr Xiang,

Thank you very much for submitting your Research Article entitled 'Integrated analysis of genome-wide association studies and 3D epigenomic characteristics reveal the BMP2 gene regulating loin muscle depth in Yorkshire pigs' to PLOS Genetics.

The manuscript was fully evaluated at the editorial level and by independent peer reviewers. The reviewers appreciated the attention to an important topic but identified some concerns that we ask you address in a revised manuscript.

We therefore ask you to modify the manuscript according to the review recommendations. Your revisions should address the specific points made by each reviewer.

Yours sincerely,

Martien Groenen, PhD

Academic Editor

PLOS Genetics

Gregory Barsh

Editor-in-Chief

PLOS Genetics

As reviewer 1 points out, you still oversell your data as providing conclusive evidence that the causal variants are located within a 10 kb region by using epigenetic data. Please tone down this statement and address the remaining comments made by the two reviewers.

Reviewer's Responses to Questions

**Comments to the Authors:**

Reviewer #1: This paper reports convincing evidence for a major QTL on pig chromosome and report functional characterization of candidate SNPs in an enhancer and the promotor of the BMP2 gene. The data are good. However, the remaining problem is that the authors still oversell their data as providing conclusive evidence that the causal variants are located within a 10 kb region by using epigenetic data. The problem here is that the QTL region is large 800 Mb and functional data can neither prove or exclude the importance of a sequence variant for a genotype-phenotype relationship, this can only be achieved by genetic analysis. For instance, a functional assay such as a luciferase assay used here can show that a sequence variant affect function but it does not prove that this effect is important for the genotype-phenotype relationship. Further, functional assay may fail to reveal a significant effect of a sequence variant because the assay does not replicate the conditions when the sequence variant is important. For instance, in this study the authors use a kidney cell line to study the possible effect of sequence variants affecting muscle development (LMD). In conclusion, I think the paper is fine if the authors correct some remaining issues described below and are more realistic what they can and cannot conclude based on these data. Hopefully, my comments below are useful.

Specific comments

L302- (Narrowing down the QTL region). I am still not convinced that these data are sufficient to narrow down the QTL region from about 800 kb to about 10 kb. The reason is that the authors have used a sparse SNP panel. There could be other sequence variants within the 800 kb region with equally strong association to phenotype as those within LD block 2. For instance, Figure 3 shows that the SNP INRA0052824 that is located about 800 kb from the top SNP still shows D’=1 (or very close to 1) to the top SNP in the 10 kb region.

Figure 4d. This figure illustrates very well that it is impossible to identify a 10 kb interval as the sole genomic region harboring causal mutations for the LMD QTL because the strong association with phenotype is over a broad region and may reflect a haplotype effect. To make this figure coherent I suggest that the authors mark their favorite region 15.65-15.75 in Fig. 4d. As far as I can see it is not at all the most strongly associated region in 4d. The authors should also explain in the legend that this plot tests for genetic association to the LMD phenotype.

Figure 5. The text referring to these experiments is a bit confusing. The authors clone 800-bp genomic regions harboring 5 candidate SNPs. The problem is apparently that these 800 bp fragments may contain other SNPs in addition to the candidate SNP. This becomes apparent in Fig. 5d when the result first is highly significant but then the authors make the specific T to G change in the T construct and then there is no significant difference. The authors conclusion on line 364-365 “that SNP 15684170 is not an important site for enhancer activity” is wrong. This is still possible but the important conclusion is that the highly significant difference between the T and G construct must be caused by another SNP in this interval, perhaps that is a candidate causal sequence variant? The authors end this section by concluding that SNP rs321846600 is the best candidate SNP. However, it is not clear if the 800 bp fragment also carries other SNPs that may be important. The authors need to sequence the four 800 bp fragments and declare which sequence differences each region contains in a Supplementary Table. If the fragment with SNP rs321846600 contains multiple sequence variants they need to mutate the candidate SNP to prove that this is underlying the difference in luciferase activity between constructs.

L432-445. I still find the authors arguments here problematic. They are mixing up genetic significance and functional significance. Epigenetic data cannot be used to narrow down a QTL region. It can be used to test the functional significance of SNPs within a QTL region. In this paper the authors have decided to focus on a 10 kb region but they have not excluded that sequence variants outside this 10 kb region but within the 800 kb region contribute to the QTL effect or even is more important than the possible effect of the variants within the 10 kb region. This paragraph needs to be modified to not be misleading to the field. In particular the sentence (The significant SNPs ..) on line 440-441 needs to be followed by something like this: “However, this does not exclude the possibility that sequence variants outside this region contribute to the LMD QTL effect in the genomic region 15.51-16.31 on SSC17.”

L468-471 There is a conflict between these two studies. I recommend that “demonstrated” is replaced with “suggested”

Minor comments

L32: delete “the”

L34: replace “the key” with “candidate”

L45-46: change to “candidate SNPs that may be functionally related to”

L51: change “major” to “candidate”

L66. Change “loci” to “locus”

L69. Change “the major” with “candidate”

L277: Change “were” to “are”

L346: Since these are predictions, I think this sentence is too strong and should read “Results suggested that these five SNPs may all disturb…”

L357 and 359: delete “different carried”

L396: change “supports the fact” with “shows”

L397: change “conservative” to “conserved”

Figure 5 and 6. Please add a few words to the legend and explain that PK15 is a porcine kidney cell line so that readers don’t have to search for this info in the M&M

Figure 5e: It is unclear what WT and MUT refer to here. It would be much better to use the nucleotides instead alleles T and C. That would make it much easier to compare the results in Fig. 5c and 5e. It may also be better to change WT and MUT to H and L in Fig. 7 because it is not clear what is WT and MUT given that these are polymorphisms segregating in pig populations.

Line 415: Add “using a 50k SNP chip” after “pigs” in order to explain that a sparse SNP panel has been used at this step.

Reviewer #3: The authors have made substantial changes to the manuscript and addressed most of my concerns from the previous review. I think it will be suitable for publication following minor changes and editing, as described below. The authors state that they have had assistance from professional editors and/or native English speakers, however, the manuscript would benefit from further editing. There are many places where the language/grammar is problematic.

Specific comments

I am not convinced that including F_ST between American Yorkshire and American Landrace breeds is a good addition to the paper. In my view, this weakens what is otherwise a rigorous study. There is no reason to expect that these two breeds will be differentiated in this region of the genome, which has been identified using GWAS within a population of Yorkshire pigs. Or at least, the authors do not provide any justification for this analysis. In terms of the Results, they show a small peak overlapping the region identified by the GWAS but this may be coincidental. There is a higher peak in the 3’ direction (about which I also wouldn’t interpret too much). As they only looked at F_ST in a small part of the genome, there is no way to tell how extreme these peaks are on the genome-wide scale. I would recommend removing this section of the paper.

Lines 278-293: I appreciate the addition of details from the previous study to which the authors refer. However, I found this section somewhat hard to follow. Are the authors saying that the Hi-C data helped them narrow down the region identified by GWAS (line 282)? I don’t see how this is the case if the TAD region is actually larger than the GWAS region.

--line 279: change “isolate” to another word (“limit”?)

--need to define “sequesters”

--line 286 (and elsewhere in the paper): need to explain in the text what is meant by “interact” and “interactions” in this context

--lines 291-293: please reword to clarify what is meant by “… SNPs are closely related to each other” and “… their target genes were expected in this TAD region”

Figure 4d. The legend for this plot needs to be changed to explain that this shows results from an association analysis (as I understand it, as described on lines 199-206?).

Line 348-365: Change “carried” to a different word (“cloned”?)

Line 389: Change “NR2C1 cannot binding” (I don’t know what this means)

Line 397: Change “conservative” to “conserved”

Line 409: Change to either “do not differ” or “are not different”

Line 422: Remove “newly developed”

Line 426: Change “verify” to “evaluate”

Line 430: Change “provides” to “identifies”

Line 472: Change to “…increased BMP2 enhancer activity…”

Line 478: Should this be “have previously been reported”?

**Have all data underlying the figures and results presented in the manuscript been provided?**

Reviewer #1: Yes

Reviewer #3: Yes

PLOS authors have the option to publish the peer review history of their article (what does this mean?). If published, this will include your full peer review and any attached files.

Reviewer #1: No

Reviewer #3: No

---

## [Editor Report · Decision Letter 2]

7 Jun 2023

Dear Dr Xiang,

We are pleased to inform you that your manuscript entitled "Integrated analysis of genome-wide association studies and 3D epigenomic characteristics reveal the BMP2 gene regulating loin muscle depth in Yorkshire pigs" has been editorially accepted for publication in PLOS Genetics. Congratulations!

Yours sincerely,

Martien Groenen, PhD

Academic Editor

PLOS Genetics

Gregory Barsh

Editor-in-Chief

PLOS Genetics

Comments from the reviewers (if applicable):

**Data Deposition**

http://datadryad.org/submit?journalID=pgenetics&manu=PGENETICS-D-22-01410R2

**Press Queries**

---

## [Editor Report · Acceptance letter]

15 Jun 2023

PGENETICS-D-22-01410R2 

Integrated analysis of genome-wide association studies and 3D epigenomic characteristics reveal the BMP2 gene regulating loin muscle depth in Yorkshire pigs 

Dear Dr Xiang, 

We are pleased to inform you that your manuscript entitled "Integrated analysis of genome-wide association studies and 3D epigenomic characteristics reveal the BMP2 gene regulating loin muscle depth in Yorkshire pigs" has been formally accepted for publication in PLOS Genetics! Your manuscript is now with our production department and you will be notified of the publication date in due course.

With kind regards,

Zsofia Freund

PLOS Genetics

On behalf of:
